# Barriers and enablers for implementation of clinical practice guidelines in maternity and neonatal settings: A rapid review

**Kalpana Raghunathan** [1]*, **Christine East** [1,2], **Kritika Poudel** [1]

1 Judith Lumley Centre, School of Nursing and Midwifery, La Trobe University, Bundoora, Victoria, Australia,
2 Mercy Health, Melbourne, Victoria, Australia

* K.Raghunathan@latrobe.edu.au

**Data Availability Statement:** Data generated and analysed for this rapid review are included in this paper and in supplementary files. The rapid review

## Abstract

### Background

Maternity and neonatal services are rapidly changing in Australia because of evolving needs of the community and patient population. Clinical practice guidelines focused on early interventions and prevention strategies can decrease risk for preventable negative health outcomes in this population. However, despite the existence of several clinical practice guidelines, their translation into practice remains problematic for healthcare services.

### Aim

To identify barriers and enablers for the implementation and adoption of clinical practice guidelines in maternal and neonatal settings.

### Methods

A rapid review was conducted according to Cochrane and World Health Organization guidelines. Systematic reviews, qualitative, quantitative and mixed-methods studies related to clinical guidelines in maternal and neonatal settings published in English Language between 2010 and 2023 meeting study eligibility criteria were identified using PubMed, Cochrane CENTRAL, EMBASE, and CINAHL databases.

### Findings

Forty-eight studies originating from 35 countries were reviewed, representing practice guidelines associated with maternal and neonatal care. Identified barriers and enablers aligned to five main themes related to the contextual level of impact: (i) healthcare system and systemic factors, (ii) patient and population, (iii) guidelines and standards, (iv) organisational capacity, and (v) health professional practice.

### Discussion and conclusion

Findings from this review shed light on the challenges and opportunities associated with introducing clinical practice guidelines in maternal and neonatal care settings.

protocol is published on Open Science Framework (https://osf.io/q47dx/).

**Funding:** The review activities were funded by Safer Care Victoria, and La Trobe University Alumni Fund, Australia. The funders had no role in study design, data collection and analysis, decision to publish, or preparation of the manuscript.

**Competing interests:** The authors have declared that no competing interests exist.

Implementation of guidelines into practice is complex, with different factors affecting their adoption and their use within healthcare settings. Addressing the multifaceted challenges associated with the implementation of clinical practice guidelines in maternal and neonatal care demands a comprehensive and collaborative strategy. Successful adoption of guidelines requires the involvement of stakeholders at all levels, supported by ongoing evaluation, feedback, and dedication to evidence-based practices.

## Introduction

Maternity and neonatal services are rapidly changing in Australia because of evolving needs of the community and patient population. Using clinical practice guidelines or good practice frameworks can promote strong quality system processes to guide service delivery focused on early interventions and prevention strategies; this can decrease risk for preventable adverse health outcomes in this population [1]. The need to reduce maternal and neonatal mortality and morbidity is critical to improving population health [2]. Moreover, demand-side interventions, such as better access to facility-based care during pregnancy and childbirth, can assist to increase uptake of critical maternal health services among women. Conversely, delays in supply-side interventions, such as the provision of timely and appropriate care by health services, contributes to poor health outcomes [2].

Clinical practice guidelines (CPG) are statements that include evidence-based recommendations for healthcare professionals about the actions to be implemented in clinical settings to optimise patient care [3]. They form the foundation for efforts to improve healthcare policy, planning, delivery, evaluation, and quality improvements [4, 5]. Guidelines are recognised as an invaluable resource to assist healthcare practitioners and clients in decision-making and may pertain to diseases or procedures [6]. They reduce variability in practice, especially in situations with multiple treatment options, or in cases of limited scientific evidence or uncertainty around the best course of action [7].

However, it is recognised that despite the development of an extensive number of CPGs, their implementation or translation into practice remains problematic and expected improvements in patient outcomes and reduction in healthcare costs remain elusive [8]. CPGs are not always applied, or applied effectively, and their adoption can be unpredictable, slow, and complex [7]. Non-adherence to guidelines may increase the potential for harm or result in the use of treatments that are unnecessary, incorrect, or not evidence based [6, 7]. Therefore, efforts to improve guideline implementation are essential to address gaps in access and quality of care and to strengthen the quality of existing services [8].

Several different CPG implementation approaches and strategies currently exist [5, 9]. However, it is recognised that varying factors can influence guideline implementation in different clinical areas, such as the socio-political context, the healthcare organisation or system, the guideline itself, as well as the individual clinician and the patient [10]. The success of guideline implementation depends on understanding barriers and facilitators for their uptake in daily practice [6, 7]. Barriers and enablers to CPG adoption may prevent or facilitate improvements in care delivery, safety and quality outcomes for individuals and organisations [6]. Therefore, identifying barriers can help organisations determine effective strategies to overcome them and improve safety and quality in clinical decision-making and minimise evidence-based practice gaps [5].

In a preliminary search for previously published reviews related to CPG implementation, the authors located two relevant articles. Fischer et al's [7] scoping review collated data up to 2015 and examined barriers and strategies in CPG implementation generally. Correa et al.'s [10] meta-review synthesised data (December 2006 to January 2018) related to barriers and facilitators that influenced CPG implementation in different clinical areas. To date, however, there has been no synthesis of factors specific to maternal and neonatal settings. Therefore, this review sought to build on this previous work through an updated and rapid review of guideline implementation focusing on maternal and neonatal service settings.

### Research aim

The aim of this structured rapid review was to identify potential factors that act as barriers and enablers for guideline implementation in maternity and neonatal settings. The research question was: What are the barriers and enablers for the implementation and adoption of clinical practice guidelines in maternal and neonatal settings related to pregnancy, labour, post-partum and neonatal services?

## Materials and methods

### Design

The rapid review method was selected for timely evidence gathering for health policy and systems, and practice decisions for a rapidly changing clinical environment [11]. This form of knowledge synthesis "accelerates the process of conducting a traditional systematic review by streamlining or omitting specific methods to produce evidence for stakeholders in a resource-efficient manner" [12]. Our review procedures were informed by: *Cochrane Rapid Reviews* interim guidance recommendations [13]; the World Health Organization's practical guide for rapid reviews [14]; and practical steps and activities for the review process [15]. Review reporting, tracking overall process and information flow was based on *PRISMA 2020 guideline for reporting systematic reviews* [16].

Aligned with the rapid review method [13], two post-hoc changes were made, which were updated in the published protocol in Open Science Framework (https://osf.io/q47dx/): (i) an updated PRISMA reporting guideline; and (ii) a decision to apply a single quality appraisal tool to screen diversity of included studies and consistency in quality assessment replacing multiple methodology-aligned tools. This research did not involve human participants, thus ethical approval was not required.

### Search strategy and procedures

An iterative process within rapid review protocol parameters assisted a robust search for literature within project time constraints [14]. An Information Specialist and MeSH© (Medical Subject Headings) terms guided the refinement of included search terms (Table 1). The Peer Review of Electronic Search Strategies (PRESS) checklist [17] and the PRISMA-S checklist for literature search reporting [18] assisted search strategy optimisation. Four databases and article reference lists were searched simultaneously for three weeks during May and June 2023. An example of the search filter from PubMed is provided in Table 1.

### Inclusion and exclusion criteria

The scope of the review was defined by PICO elements (i.e., population, intervention, comparator, outcome), as well as setting, timeframe, and study design, which are recommended for a rapid review [14] (Table 1). Peer-reviewed, systematic reviews, qualitative, quantitative and

**Table 1. Study eligibility and search terms.**

| Eligibility criteria and search | | Search terms |
|---|---|---|
| Population (Concept 1) | Health professional, clinicians | MeSH terms: Health personnel and Maternity Hospitals, Pregnancy, Maternal health service, Infant, Newborn, Midwifery, Delivery obstetric, Prenatal care |
| Intervention (Concept 2) | Clinical recommendations, policies, clinical guideline, evidence based, policy | MeSH terms: Clinical recommendations, Policies, Clinical guidelines, Evidence based |
| Comparator (Concept 3) | Barriers, enablers or facilitators | MeSH terms: Barrier*, Enable*, Facilitat*, Strateg* |
| Outcome (Concept 4) | Implementation, adoption, uptake | MeSH terms: Adopt*, Uptake, Compliance, Accept*, Conform, Approv*, Adherence, Apply*, Implement* |
| Study design Publication | Systematic reviews, qualitative, quantitative and mixed methods studies Peer reviewed, English Language between January 2010 and May 2023 | |
| Databases searched | PubMed, Cochrane CENTRAL, EMBASE, and CINAHL | |
| Search filter in PubMed | (guideline*[Title/Abstract] OR guidance*[Title/Abstract] OR clinical protocol*[Title/Abstract]) AND (strateg*[Title/Abstract] OR barrier*[Title/Abstract]) AND implement*[Title/Abstract] AND (compliance[Title/Abstract] OR accept*[Title/Abstract] OR conform*[Title/Abstract] OR approv*[Title/Abstract] OR adherence[Title/Abstract]) AND (pregnancy**[Title/Abstract] OR Infant*[Title/Abstract] OR newborn*[Title/Abstract] OR neonatal*[Title/Abstract] OR labour*[Title/Abstract] OR obstetric*[Title/Abstract] OR gynaecology*[Title/Abstract] OR postpartum*[Title/Abstract] OR maternity*[Title/Abstract]) | |

mixed-methods studies published in English Language between January 2010 and 30 May 2023 were included. No geographical limits were applied. Project reports, unpublished and grey literature, study protocols without results, and conference abstracts were excluded.

## Study selection and screening

Screening and study selection was undertaken in Covidence (Covidence systematic review software, Veritas Health Innovation, Melbourne, Australia. Available at www.covidence.org). Two authors independently recorded their decisions at each stage (titles/abstracts and full texts), with conflicts resolved by the third as required. Fig 1 presents PRISMA flow chart and study selection.

## Quality assessment

Risk of bias and quality appraisal of studies was assessed with the Quality Assessment with Diverse Studies (QuADS) tool which is an appraisal tool for methodological and reporting quality in systematic reviews of mixed- or multi-method studies [19]. The QuADS tool involved a scoring schema from 0 to 3 based on 13 criteria (maximum score of 39). Two authors independently evaluated the same five studies for inter-rater reliability, with discussion of quality assessment and resolution of scoring discrepancies through moderation, resulting in similar final scores. One author completed quality assessment for the remaining studies. Overall scores as percentages of the criteria were used to compare and critique the articles. Studies were not excluded based on quality appraisal.

## Data extraction and synthesis

Relevant data were extracted from included documents, summarising the overall findings and forming conclusions to answer the research question [15]. Data were extracted by the first author, then independently verified by two other authors. Ritchie and Spencer's [20] framework approach provided a structured method suited to qualitative data, and document analysis was used to identify barriers and enablers to guideline implementation and theme development. A general inductive and deductive approach was used to distil and condense extracted informational text (data) from the literature to generate themes [21]. This was managed

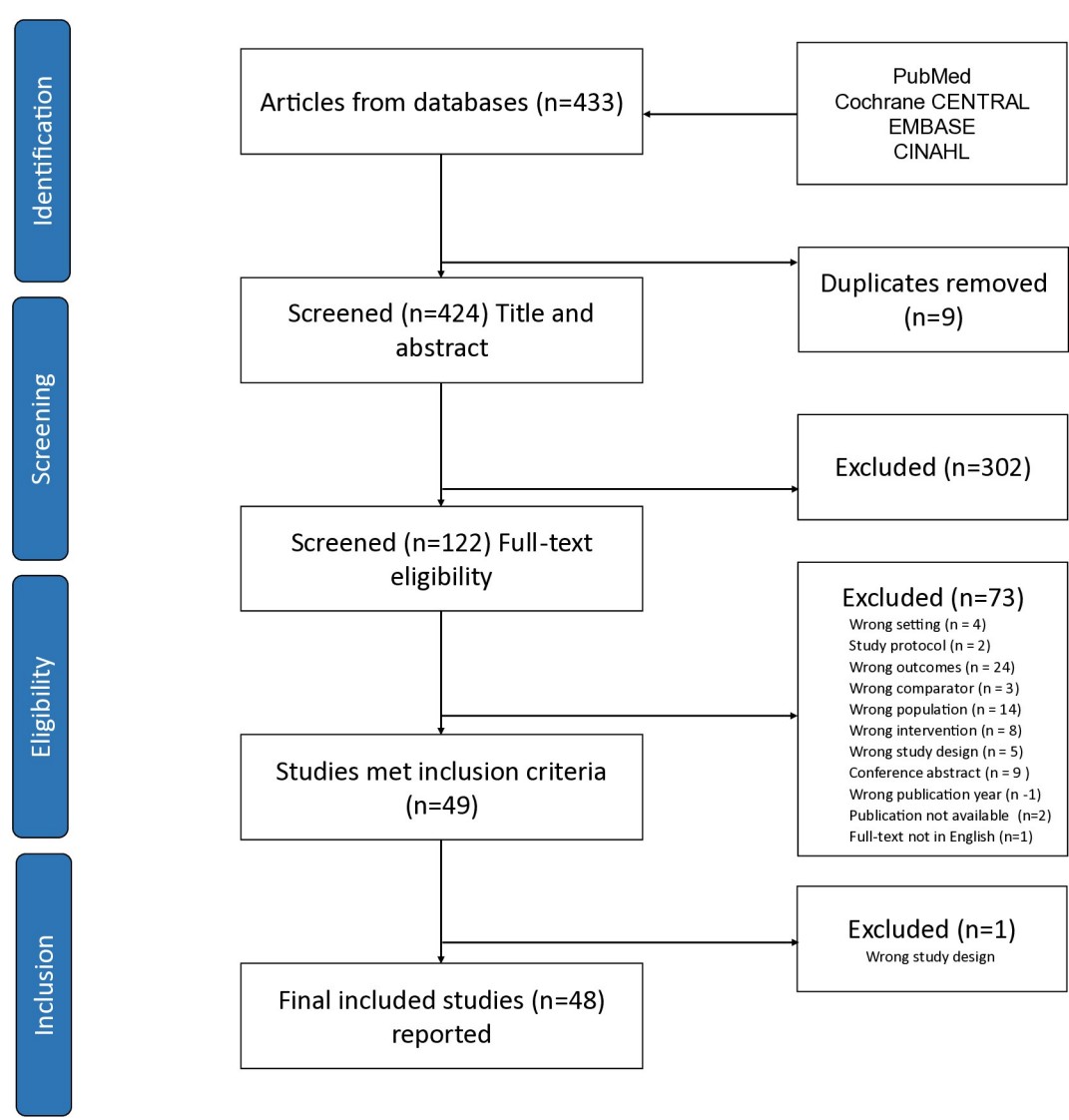

**Fig 1. PRISMA flow chart and study selection.**

through five iterative steps [20]: (i) familiarisation with the literature, (ii) generating overarching conceptual constructs, (iii) indexing and sorting coded data and concepts, (iv) summarising initial themes, and (v) mapping and interpreting data and final themes. Table 2 provides examples of theme development. Data were managed in Microsoft Word and Excel. Findings are presented in a descriptive narrative form as study characteristics and key themes related to barriers and enablers.

## Results

### Study characteristics

A purposive database search and full-text screening yielded 49 papers for inclusion, of which a single paper was excluded by the reviewers jointly during data extraction since it did not meet the study design criterion (Fig 1). S1 File lists studies screened. In total, 48 papers were included for analysis and reporting, summarised in Table 3. (Also see S2 and S3 Files). Reports

**Table 2. Example of coding and theme development.**

| | Extracted data | Category-sub theme (Coding) | Level of impact or application (main theme construct) |
|---|---|---|---|
| BARRIERS | 'Cost factors—booking fee and cost of travel to hospital (against a background of poverty, low resources, long distance to travel)' | Resource limitations | Women and community-based challenges (patient and population) |
| | 'Different guidelines used to inform practice (national and international)' | Variability in guidelines | Guidelines and standards related challenges |
| | 'An overburdened national health system' | Resource limitations-low resource settings | Health care system or systemic challenges |
| | 'Resistance to change on the part of health professionals' | Resistance to change | Health professional related challenges |
| | 'Lack of alert-reminder design /interface in electronic records' | Lack of decision support systems and processes | Organisation (healthcare provider) related challenges |
| ENABLERS | 'Educating nurses and midwives about the continued monitoring and reporting of unremitting symptoms in the postpartum period to minimize any possible complications' | Continued education and training about monitoring and reporting data | Health professional practise |
| | 'More consistent and evidence-based guidelines for the paediatric setting could improve confidence in recommendations and hence compliance with them' | Consistent and evidence-based guidelines to improve confidence in recommendations | Guidelines and standards development |
| | 'Automated orders as part of care maps and electronically generated messages to facilitate adherence' | Automated care pathways and electronic reminders | Organisation/facility initiatives |
| | 'Simplify test and make it easier for woman' | Redesign screening and tests to improve patient uptake | Health care system |
| | 'Patient education and instructional handouts' | Patient education | Women and community (Patient and population) |

originated from different countries, including cross-regional research collaborations. More than half represented four countries, namely, Australia (n = 9), United States (US) (n = 6), United Kingdom (UK) (n = 6) and New Zealand (n = 4) (Table 3). Twenty-seven studies were published since 2018. Study methodology included quantitative (n = 31), qualitative (n = 7), mixed methods (n = 4), and systematic reviews (n = 6). Most studies considered guideline use in hospital settings (n = 35) and related to primarily medical, nursing and midwifery health professionals involved in maternity and neonatal care. Quality assessment for methodological and reporting quality indicated that all included studies individually scored more than 75% against the QuADS tool criteria (S4 File), which is acceptable evidence since different types of study designs were included in the review (Table 3).

## Barriers and enablers to guideline implementation

Key barriers and enablers related to guideline implementation in maternity and neonatal settings are summarised and grouped under five main themes, based on contextual level of impact or application, with some overlap between themes acknowledged (Tables 4 and 5). Also see S5 File.

## Healthcare system factors

Healthcare system factors concern structural factors, including the scope of services, support infrastructure, and the socio-economic and policy environment of its geographical location. A key barrier impeding guideline uptake and adherence, particularly in low-income settings, were the lack of resources and economic factors which impacted scope of maternal services and quality of care as illustrated on Table 4. These affect both the healthcare system and the population (see patient and population factors). Other barriers related to models of care, such

**Table 3. Summary of studies reviewed.**

| Author, year | Study origin/ country | Study type | Clinical practice guideline/policy/ recommendation | Quality appraisal percentage of criteria met |
|---|---|---|---|---|
| Akuma 2012 | United Kingdom | Quantitative | Guideline by Association of Paediatric Anaesthetists of Great Britain and Northern Ireland 2008. | 82% |
| Albouy-Llaty et al. 2012 | France | Quantitative | Perinatal Group B streptococcus (GBS) screening guidelines. | 85% |
| Alja'freh and Abu-Shaikha 2021 | Jordan | Quantitative | Clinical practice guidelines of Hypertensive Disorders of Pregnancy. | 90% |
| Alsweiler et al. 2020 | New Zealand | Quantitative | Neonatal hypoglycaemia guideline. | 82% |
| Breakell 2018 | United Kingdom | Quantitative | National Institute of Health and Care Excellence bronchiolitis guideline. | 97% |
| Brower et al. 2019 | United States | Quantitative | Local guideline recommendation for treatment of Herpes Simplex Virus and use of acyclovir in neonates and infants. | 95% |
| Brozanski et al. 2020 | United States | Quantitative | Perioperative euthermia clinical practice recommendations. | 92% |
| David et al. 2021 | Zimbabwe | Mixed methods study | Prenatal care pathways for pregnant women. | 85% |
| da Silva Carvalho et al. 2021 | Brazil | Mixed methods study | National Clinical Guidelines for Care in Normal Birth. | 85% |
| de Oliveira Carvalho 2013 | Brazil | Qualitative | Maternal breastfeeding clinical management of lactation and the orientation/techniques to prevent early difficulties during breastfeeding. | 77% |
| Doherty et al. 2022 | Australia | Quantitative | Model of care addressing alcohol consumption based on systematic review of evidence, international and Australian clinical guidelines. | 97% |
| Eldh 2016 | Sweden | Qualitative | Guideline for peripheral venous catheters management in paediatric care. | 95% |
| Gkentzi et al. 2017 | United Kingdom | Systematic review | Recommendations for national immunisation program- antenatal vaccination against pertussis. | 90% |
| Gu et al. 2020 | China | Quantitative | Guideline of Enteral Nutrition for Infants with Congenital Heart Disease. | 95% |
| Haskell et al. 2021 | New Zealand and Australia | Mixed methods study | Targeted theory-informed interventions to improve bronchiolitis management in acute paediatric setting. | 100% |
| Kebaya et al. 2018 | Kenya | Quantitative | Evidence-based criteria regarding newborn resuscitation in maternity units. | 100% |
| Langley et al. 2015 | Canada | Quantitative | Canadian Nosocomial Infection Surveillance Program for Methicillin-Resistant Staphylococcus aureus transmission in paediatric health care facilities. | 87% |
| Laubscher et al. 2013 | Switzerland | Quantitative | Swiss guidelines to prevent paediatric vitamin K deficiency bleeding. | 77% |
| Luitjes et al. 2018 | Netherlands | Quantitative | Obstetric guidelines on the management of hypertension in pregnancy. | 97% |
| Lyngstad et al. 2022 | Norway | Quantitative | Guidelines for pain assessment and management and increased parental involvement in single-family room NICU. | 97% |
| Mohan et al. 2023 | United States | Quantitative | American Academy of Paediatrics guideline recommendations for intravenous immunoglobulin in infants with haemolytic disease. | 87% |
| Moore et al. 2020 | Canada | Qualitative | Guideline supporting shared decision making for extreme preterm birth. | 92% |
| Muhumuza et al. 2015 | Uganda | Quantitative | Paediatric special care unit Hand hygiene to reduce transmission of health care worker-associated pathogens. | 95% |
| Muirhead and Kynoch 2019 | Australia | Quantitative | Evidence-based clinical guideline for the management of neonatal pain published by the Australian New Zealand Neonatal Network. | 100% |
| Nair et al. 2014 | United Kingdom | Systematic review | Quality of care (World Health organisation WHO framework) for pregnant women, newborns and children. | 92% |
| Nkamba et al. 2017 | Zambia and Democratic Republic of Congo | Qualitative | Antenatal screening and treatment during pregnancy. | 92% |
| O'Loughlin et al. 2021 | Lao | Quantitative | Integrated management of neonatal and childhood illness guidelines-national strategy. | 87% |

*(Continued)*

**Table 3.** (Continued)

| Author, year | Study origin/ country | Study type | Clinical practice guideline/policy/ recommendation | Quality appraisal percentage of criteria met |
|---|---|---|---|---|
| Olsen et al. 2018 | United States | Quantitative | Nutritional guidelines for premature infants. | 95% |
| Page et al. 2017 | Australia | Qualitative | Nutrition guidelines for infants who weigh <1500 gms (preterm birth). | 85% |
| Pangerl et al. 2021 | Australia | Systematic review | GBS Screening Guidelines in Pregnancy. | 87% |
| Pauws et al. 2017 | Netherlands | Quantitative | Implementation of manual oxygen titration guideline for pre-term infants. | 85% |
| Pricilla et al. 2018 | Kenya | Quantitative | WHO's Prevention of mother to child transmission of *Human immunodeficiency virus* (HIV) treatment guidelines. | 92% |
| Rousseau et al. 2020 | France | Mixed methods | National guidelines for obstetrics. | 92% |
| Ryan et al. 2020 | United States | Systematic review | Postpartum haemorrhage clinical guidelines, policy and management of Obstetric haemorrhage prevention. | 100% |
| Sharma et al. 2021 | Norway | Qualitative | Lifestyle-changes guidelines essential for preventing diabetes post-gestational diabetes mellitus. | 97% |
| Silva et al. 2013 | Brazil | Quantitative | Guidelines for GBS prenatal screening. | 85% |
| Skåre et al. 2018 | Norway | Quantitative | Neonatal resuscitations criteria. | 92% |
| Smith et al. 2017 | United Kingdom | Qualitative | Maternal death surveillance and response and Maternal Death Review systems. | 92% |
| Snelgrove-Clarke et al. 2015 | Canada | Quantitative | Fetal health surveillance guideline in clinical practice. | 92% |
| Stokes et al. 2016 | New Zealand | Systematic review | Guidelines to improve obstetric care practice. | 92% |
| Sundercombe et al. 2014 | Australia and New Zealand | Quantitative | Postnatal ward neonatal hypoglycaemia guidelines and UNICEF UK Baby Friendly Initiative recommendations. | 92% |
| Telfer et al. 2021 | United States | Quantitative | Evidence-based bundle to reduce early labour admissions and labour management guidelines associated with decreased caesarean birth. | 100% |
| Trevisanuto et al. 2015 | Vietnam | Quantitative Survey | International guidelines for neonatal resuscitation. | 95% |
| Trollope et al. 2018 | New Zealand | Quantitative | Maternity clinical practice guidelines developed by 'National Women's Health'. | 90% |
| Turan et al. 2012 | Kenya | Quantitative | Antenatal Care Integration in Pregnancy. | 90% |
| Warren 2011 | Australia | Quantitative | Protocol for the prevention and management of extravasation injuries in the neonatal intensive care. | 92% |
| Wilkinson et al. 2017 | Australia | Quantitative | Clinical guidelines regarding weight management in pregnancy- best practice delivery of care to pregnant women regarding gestational weight gain. | 87% |
| Zahroh et al. 2022 | Australia and Switzerland | Systematic review | Use of antenatal corticosteroids, tocolytics, magnesium sulphate, and antibiotics to improve preterm birth management. | 92% |

as fragmented maternity care that is not woman-centred. This affected the quality and consistency of services, such as provision of routine screening or a consistent schedule of pregnancy care visits [22, 23] and information flow across services at district and central levels, including limited access to current guidelines [24, 25].

Conversely, structural factors such as a well-planned healthcare system, policy priorities with resources and investment for maternal health facilitated practice change and guideline adoption at the system level (Table 5). For instance, national and local level government commitment and investment into public health initiatives were pivotal to improving maternity services and quality of care, particularly in low-resource settings [23–29]. Multiple studies (n = 13) (Table 5) identified that mechanisms for accountability through governance of healthcare practitioner regulators, incentivisation for practice change and positive outcomes

**Table 4. Summary of barriers related to guideline implementation and adoption to related to maternal and neonatal care.**

| Healthcare system factors | |
|---|---|
| Lack of resources: predominantly low- and middle-income countries (LMICs) | Low resource settings and LMIC's lack of capacity to train and update their staff in management of maternal and neonatal care [24, 26, 28, 30]. Lack of financial capacity, lack of equipment and poor quality of facilities for maternal services [22]. Lack of critical clinic and laboratory supplies and essential medicines impacted service delivery, quality of care and practice change [29, 36, 62]. Lack of human resources including lack of trained or qualified health professionals to provide maternity/neonatal care, with flow-on affect for service delivery [36, 45]. Resource limitations also compounded by environmental factors and geographical limitations; women may be unable to travel to health centres, unable to pay for service; lack of access to clean water affecting quality of maternal and neonatal care at a time of vulnerability to infection [49]. |
| Models of care | Lack of woman-centred and comprehensive care, and fragmentation of maternity care in the public health system [22, 23]. Haphazard nature of offering routine screening and consistency of antenatal visits [27, 28, 33]. Challenges to introduce interventions (including preterm birth prevention and management guidelines) in the context of existing substandard intrapartum, birth and newborn care [36]. |
| Poor communication and coordination | Poor information flow between district and central committees, and across the health system with healthcare providers being unaware or not able to access these guidelines [24, 25]. For example, Maternal Death Surveillance and Response requires government commitment for training, maternal death classification and formulating recommendations [24]. |
| Macro-micro level factors | Contextual factors and internal and external environment of the organisation [53]. Political and economic environment, organisational status and culture, regulatory frameworks, resource allocation and system-level support [23, 24, 36, 37, 46]. |
| Overburdened health system | Overburdened national health system, lack of resources, materials and staff shortages, poor accessibility (functional), supply chain bottlenecks [22, 23, 26]. |
| Conflicting priorities and lack of policies | Conflicting healthcare investment priorities, resistance from government, political disinterest [22]. Lack of targeted healthcare policies [22, 24, 27, 49]. |
| Patient and population (Women and community) | |
| Costs and financial resource limitations | Cost effectiveness and acceptability of screening, treatments or medications recommended by guidelines ultimately affecting patient outcomes and quality of care [28, 29, 31, 32]. Lack of access to health insurance or financial constraints hindered women's abilities to follow guideline recommendations, especially when expensive treatments or medications are recommended [23, 26–28, 50, 63]. Financial constraints were more pronounced for low-resource settings with geographic disparities that made access and follow-up care difficult for woman [23, 29]. |
| Social and cultural influences | Reliance on alternative faith-based care [26, 28], and social vulnerabilities [50, 63]. Women's real-life constraints [33]. Stigma associated with sexually transmitted disease and need for partner consent to seek healthcare [23]. |

(*Continued*)

**Table 4.** (Continued)

| **Healthcare system factors** | |
|---|---|
| Lack of health literacy | Low level of health literacy and awareness of services; lack of knowledge about consequences and intervention benefits among the women (and parents) [23, 27, 28, 34, 36, 50]. |
| Patient factors (belief, preferences, practices) | Women's belief and personal preferences for healthcare and fear of side effects and treatment legitimacy [23, 27, 33, 36, 50, 64]. |
| Lack of stakeholder involvement | Lack of patient and population involvement and joint decision-making opportunities about their care [27]. |
| **Guidelines and standards** | |
| Multiple or different guidelines in use | Presence of different international and national guidelines [29, 30, 44]. <br> Different clinical protocols and conflicting recommendations across health settings causing practice variations and inconsistencies [23, 41–43]. |
| Guideline availability and access | Lack of guideline availability; difficult to locate; lack of accessibility at the point of care for decision support [44–46]. |
| Complexity and guideline applicability | Complexity, lack of clarity and length of guidelines [43, 46]. <br> Lack of contextualisation and relevance; did not fully account for local variations aligned to resources, or diversity of patient populations or the operating healthcare environment [23, 43, 44, 46, 47, 60]. |
| Variability in guideline development and quality | Lack of a rigorous development process; lack of sufficient evidence-based recommendations [43, 44, 47]. <br> Oversimplified; credibility and applicability concerns undermining clinicians' confidence in recommendations [30, 43, 44, 46]. |
| Lack of clear benchmarks or standards for practice | Inconsistent clinical guidelines and protocols used to guide practice decisions [36]. <br> Different guidelines used to inform practice (national and international) [27, 35, 44]. |
| Contextual implementation challenges | Lack of planning and insufficient impact monitoring systems [24]. <br> Delays and changes in services resulting in multiple guideline changes over the implementation timeline affecting desired outcomes [29]. |
| **Organisational capacity (healthcare organisation, service, or facility level)** | |
| Resource constraints | Lack of necessary resources (financial, human, and technological) to support implementation of guidelines at the service level; limited access to essential equipment, technology updates and tools; inability to provide needed training to support health professionals [22, 28–30, 35, 45, 47, 49, 50]. |
| Practice variations in organisations | Different protocols and practices at individual centres or facilities within the health service [24, 28, 30, 35, 36, 46, 48, 52, 53]. <br> Variations in referral practice; variable practice opinions of clinicians [35]. <br> Lack of consensus about interventions and measures to apply in settings [42]. |
| Workflow organisation | Inefficient point of care workflow processes, paper-based documentation rather than electronic tracking systems and alerts [28]. <br> Lack of triggers or reminders as decision support aids integrated with routine clinical workflow into the electronic medical record system for clinicians at the point of care [28, 51]. <br> Poorly designed electronic decision alerts at point of care [65]. <br> Outdated diagnostic tools and algorithms [39]. <br> Lack of automated communication reminders for pregnant woman to adhere to scheduled appointments [28]. |

(*Continued*)

**Table 4.** (Continued)

| Healthcare system factors | |
| --- | --- |
| Environmental and contextual factors | Organisational status of maternal health services (public-private, non-university, small centres) and resources available to support guidelines implementation [29, 46]. Location-rural maternal centres spread geographically; frequent reassignment of maternal staff between services; diversity of facilities and populations served; large distances to travel and difficulties with follow-up maternity care; inadequate transportation systems to deliver supplies [29]. |
| Organisations' capabilities | Shortage of well-trained healthcare workers and knowledge discrepancies among different levels of staff [24, 52]. Lack of clinical leadership [25, 27]. Lack of quality improvement initiatives and systems for monitoring guideline adherence and providing feedback to staff [24, 25, 29]. Variation in quality improvement capabilities across centres and culture [53]. |
| Lack of team communication and collaboration | Limited communication and collaboration among different healthcare disciplines [24, 25, 37, 45, 51, 66]. Professional indifference to innovative strategies [51]. Traditional medical hierarchies; lack of stakeholder consensus; blaming exercise culture; poor communication of audit meeting feedback to clinicians [25]. |
| Inadequate dissemination of guidelines | Inadequate communication or restricted dissemination of guidelines [22, 25, 27, 46]. |
| Quality of data and data management systems | Lack off or missing data; poor quality of data collection [24, 25, 39]. Untrained and inexperienced staff and un-motivated data collection [24, 25]. Inadequate response monitoring and data management systems in use [29]. |
| Guideline implementation challenges | Lack of planning around implementation [24]. Lack of guideline adherence and monitoring systems in place [24, 25, 29, 31]. |
| **Health professional practice (clinicians)** | |
| Lack of guidelines awareness | Lack of current guideline awareness; minimal familairity with guideline content or recommendations for practice; current knowledge and skills deficit [23, 26, 32, 35, 37, 41, 45–47, 49, 57, 60, 62, 66]. |
| Lack of professional motivation and engagement | Professional indifference; lack of motivation (without incentives) to attend training [22, 25, 45, 51, 58, 66]. |
| Resistance to change | Health professional resistance to change; loss of autonmy [22, 38, 44, 46, 48, 52, 66]. |
| Health workers' attributes and attitudes | Variations in guideline adherence between disciplines [59]. Variations in individual practice [35, 38, 46]. Variable knowledge and health workers' practice knowledge and skill gaps [24, 26, 27, 30, 32, 36, 41, 44, 45, 48, 51, 52, 60, 66, 67]. Lack of awareness of the degree of noncompliance [42, 60]. Personal beliefs and attitudes (beyond or outside their scope of duties); longstanding or entrenched practices; clinician perceptions [36, 44–46, 48, 49, 51, 60, 67]. |
| Lack of interdisciplinary collaboration | Poor collaboration between health disciplines and units or clincial settings [27, 45, 66]. Traditional health profession hierarchies [25]. |
| Time constraints and workload | Time constraints; heavy workload; busy units [35, 46, 60–62, 66]. |
| Poor quality of reporting | Poor recording; inaccurate and inconsistent reporting; poor documentation quality [24, 25, 28, 60]. |
| Lack of education and training about guidelines | Lack of education and training about guidelines and updates [22–24, 30, 66]. |

**Table 5. Summary of enablers or facilitators for guideline implementation and adoption related to maternal and neonatal care.**

| Healthcare system factors | |
|---|---|
| Healthcare system structure, services, and delivery | Government commitment, clinical leadership, cost effective healthcare services [23–29]. |
| Healthcare priorities, public health resources, and investment | Healthcare investment and public health initiatives to improve services and quality of care in low-resource settings [23–29]. |
| Practice, regulation, standards, incentivisation | Standardisation of guideline recommendations across maternity services [30].<br>National guidelines for antenatal care to improve care within primary care clinic settings [23].<br>Mechanisms for accountability and mandatory practice standards, monitoring of performance indicators for organisations and health professionals [22, 24, 31, 44].<br>Reimbursement structures for healthcare providers, renumeration and rewards for practice change and positive outcomes to promote guideline adherence and uptake by healthcare providers [30, 44, 50, 66]. |
| **Patient and population (Women and community)** | |
| Patient attributes and experience | Patient knowledge and awareness of services and what to expect [36, 49, 50].<br>Positive experiences and perceptions of care model [63].<br>Involvement in healthcare decision-making and engagement by healthcare services [27].<br>Recognition of importance of early treatment and adherence to planned care [23].<br>Established trust and relationship between woman and healthcare providers, along with support, autonomy and empowerment [36]. |
| Resources and support for patients | Established trust and relationship between woman and healthcare providers, along with support, autonomy and empowerment [36].<br>Targeted educational material and sufficient support for women with decision making and treatment adherence [23, 27, 28, 34–38].<br>Improved monitoring and support for medication adherence and retention in care [39].<br>Multilingual literature for pregnant women to assist with decision making [37].<br>Electronic or automated booking, referral tracking and appointment reminder systems for pregnant women [28]. |
| **Guidelines and standards** | |
| Guideline standardisation and quality | Development of explicit local-institution protocol based on clinical criteria and recommendations [32, 60].<br>Guideline standardisation and quality of evidence on which the actionable recommendations were based [27, 30, 44, 45, 48].<br>Well-developed standardised clinical practice benchmarks to improve confidence in the guideline recommendations and a practice change [44, 45].<br>Clear unit policies and clinical indicators assisting decision-making specific to the practice environment; relevant and useful decision-making aid to improve patient outcomes [30, 48]. |
| Design, accessibility guideline usability | Point of care availability of an evidence-based guideline and interventions targeted at provider engagement [47, 68].<br>Develop point of care high-impact visual decision support tools and checklists to drive guideline adherence [38, 46, 56].<br>Integration of guideline-based checklists and screening tools into the routine workflow; easy digital access through electronic medical records at the point of care [38, 47, 56, 60]. |
| Guideline development and responsibility | Strategies improving involvement (engagement) and role of staff in guideline development process; greater responsibility in its promotion within the organisation [37, 41, 45, 56]. |
| **Organisational capacity (healthcare organisation, service, or facility level)** | |

*(Continued)*

**Table 5.** (Continued)

| | |
|---|---|
| Quality improvement initiatives | Health professionals' practice and guideline adherence monitoring and periodic audits [23, 25–27, 29, 31, 36–38, 41, 43, 51–53, 57, 61, 66, 69, 70].<br>Strategic well-planned and targeted action plan and implementation activities [23–27, 37–39, 44, 47, 52, 53, 55, 57, 58, 60–63, 68].<br>Use of implementation frameworks:<br>Plan-Do-Study-Act (PDSA) cycles [38, 53, 55, 56].<br>Joanna Briggs Institute (JBI) Practical Application of Clinical Evidence System (PACES) and Getting Research into Practice (GRiP) audit and feedback tool [49, 52, 66].<br>Theoretical Domains Framework (TDF) to identify barriers [57].<br>Theory of change to guide intervention activities [26]. |
| Organisational support and stakeholder engagement | Championing and resources to support change [23, 26, 27, 35, 45, 47, 49, 53, 55–57, 60, 61, 63, 66].<br>Stakeholder involvement in guideline development and implementation process [26, 27, 35, 36, 38, 44, 45, 48, 52, 53, 55–58, 61, 63, 66]. |
| Dissemination of information and education | Dissemination of guideline content; education and training updates for clinicians; promoting awareness of guideline recommendations [22, 23, 25–27, 29, 30, 32, 35–37, 40, 45–47, 49, 52, 53, 55, 56, 58, 60, 62, 66, 68]. |
| Work design and decision support mechanisms | Integration of guideline practice into the clinical environment; integration of decision-support mechanisms into routine workflow; integration of guideline into electronic medical records in patient charts for ease of access; development of automated, just in time reminders and triggers to support clinical management according to recommendations [24, 27–29, 36–38, 49, 51, 53, 55, 56, 60, 66, 68, 69]. |
| Health data management systems | Continuous compliance monitoring and data collection with shared data repository [24, 29, 53].<br>Electronic data management and health information systems [24, 25, 45]. |
| **Health professional practice (clinicians)** | |
| Education and training | Education and training; feedback and increasing awareness of guideline recommendations to change health professionals' practice [22, 24, 27, 29, 32, 34–37, 40, 41, 45, 49, 52, 55, 56, 60–62, 66, 67, 70]. |
| Interdisciplinary engagement | Interdisciplinary team communication and guideline education; team learning; interdisciplinary collaboration to improve care quality [25, 27, 32, 36, 53, 55, 60, 66]. |
| Practice autonomy | Providers permitted to deviate from the guidelines with proper documentation in the medical record [56]. |
| Professional involvement in planning and service delivery | Clinicians' role in guideline promotion; professional identify and involvement in planning and service delivery [24, 25, 27, 36, 41, 45, 55, 57, 61, 66]. |
| Health professionals' positive behaviour change | Health professionals' awareness of guideline recommendations; positive perceptions about guideline usefulness; beliefs and values for positive patient outcomes; positive attitude and commitment to practice change [35, 44, 45, 47, 57, 58, 63, 66, 67]. |
| Established practice standards | Clarity of information; clear policies. expectations and practice standards for health professionals; clear clinical indicators impacting decision-making [27, 37, 41, 42, 44, 45, 48, 52, 56, 60]. |

facilitated guideline adherence and uptake by healthcare providers. Guideline uptake and adherence were also facilitated by centralised and standardised pregnancy care guidelines at state or national levels [23, 30].

## Patient and population factors

Patient and population factors pertained to acceptance of guideline recommendations associated with maternity care among childbearing women, their support persons (and community),

as well as parents of neonates as recipients of healthcare. Barriers included lack of influence or direct involvement in guideline development and shared decision-making, compounded by women's previous poor experiences with healthcare systems, low health literacy, poverty, cultural and social influences and scepticism about the value and legitimacy of guidelines (Table 4). In some cultures, there was reliance on alternative faith-based care, or women could not attend a primary care clinic without the family's or partner's approval [26, 28]. Women's willingness and ability to access pregnancy care were associated with service fees, a lack of health insurance, travel costs, and long distances to maternal health centres [23, 28]. Inefficiencies in healthcare system factors affected women's perceptions of the cost effectiveness and acceptability of attending pregnancy visits, screening, treatments or medications recommended by guidelines [22, 23, 28, 29, 31–33]. These barriers ultimately affected maternal and neonatal health outcomes and quality of care [28, 29, 31, 32].

Conversely, as illustrated on Table 5, patient attributes and positive experiences and engagement with healthcare services facilitated guideline implementation. Elements such as establishing trust and rapport between women and healthcare providers, along with financial support to attend appointments [23, 27, 28, 34–38], promotion of autonomy and empowerment [36] and the availability of targeted and multilingual resources helped women with decision-making and care plan adherence [23, 27, 28, 34–38]. For example, pregnant women's participation and decision-making within a program of antiretroviral medication was enhanced by improved monitoring, multilingual literature and support [37, 39]. This was critical to achieve national and global public health targets for antiretroviral therapy coverage in prevention of mother-to-child transmission of human immunodeficiency virus (HIV) among African women [39]. Similarly, the availability of multilingual and targeted educational material influenced uptake of screening for Group B Streptococcus (GBS) [31, 37, 40].

## Guidelines and standards

Barriers associated with clinical guidelines themselves included a lack of, or poor: standardisation, quality, accessibility, feasibility of implementation across settings, and engagement with end users in their development (Table 4). Inconsistent or conflicting guidelines (i.e. that recommended different processes for the same condition) were unlikely to be followed [23, 29, 30, 41–44] and resulted in practice variations among clinicians and inconsistencies in care delivery [41–43]. For example, survey of maternity hospitals in Vietnam, identified the use of different international and national protocols for neonatal resuscitation among public maternity hospitals and the provincial and district level hospitals [30].

Guidelines that were difficult to access via the internet/intranet [44–46], considered to be too complex, or were inappropriate in the context of available resources or population diversity were also an issue [23, 43, 46, 47]. For example, the most commonly cited barrier to using a set of New Zealand maternity guidelines was locating them on the internet [44]. Guideline complexity, lack of clarity and lack of alignment with 'common sense' and the local context were cited as reasons for obstetricians' low adherence to national guidelines for prevention of preterm birth in France [46]. Also, clinicians' confidence to implement guidelines was undermined in the absence of a rigorous development process and lack of sufficient evidence-based recommendations [30, 43, 44].

By contrast, enablers at the guideline-level included quality, design, accessibility and usability of guidelines, clinician engagement and rigorous guideline development (Table 5). For instance, clinicians were more likely to view fetal surveillance and neonatal resuscitation guidelines as worthwhile and useful decision-making aids if there were clear unit policies and clinical indicators specific to the practice environment [30, 48]. Also, well-developed,

standardised clinical practice benchmarks were important to improve confidence in guideline recommendations and supporting practice change [44, 45].

## Organisational capacity

Guideline adoption is influenced by organisational factors at the facility or health centre level. These factors may be financial, human, technological and influenced by the presence or absence of robust quality improvement initiatives, implementation strategies, service culture and support, educational structures, data management systems and other work designs.

Various types of resource constraints at the service level impacted guideline adoption in practice (Table 4). For example, logistical and financial barriers limited the provision of supplies for pregnancy care in some countries [28, 29, 49]. Compliance with the national antenatal pertussis immunisation programs in some countries were affected by high vaccine costs, storage, and inventory requirements [50]. Other contextual examples include: lack of equipment affecting the provision of critical care for extremely premature infants [47]; lack of readily available essential equipment such as weigh scales causing practice discrepancies and poor milestone montioring of women's gestational weight gain [35]; and inadequate human resources leading to a shortage of maternity and neonatal staff [28, 45].

Different work design issues within the organisation or health centre also impinged guideline adherence (Table 4). For instance, paper-based, rather than electronic medical records, and lack of reminders or in-built decision support aids and timely guideline alerts at point of care, affected guideline adherence and the ability to monitor quality of maternity and neonatal care [28, 51]. Environmental or contextual elements such as the organisation's location and size also affected its ability to implement and support guideline use without the necessary mechanisms to bring about practice change [29, 46].

Deficits in organisational capabilities were linked to lack of human resources. These included shortages of well-trained healthcare workers and knowledge discrepancies among different levels of maternity staff affecting consistency and quality of care [24, 52]. Poor workplace culture additionally hampered guideline adherence. Examples included: lack of clinical leadership within maternity care [25, 27]; lack of quality improvement initiatives and lack of systems for monitoring guideline adherence and providing feedback to staff [24, 25, 29, 53].

Insufficient communication and collaboration between disciplines (i.e. doctors, midwives, nurses) was also a problem for guideline uptake (Table 4). For instance, Gu et al. [45] found that lack of multidisciplinary cooperation and working processes, as well as limited communication between doctors and nurses about infant nutrition, impeded efforts to implement a guideline for enteral nutrition risk screening for infants with congenital heart disease at a large tertiary hospital in China, with the nurses perceiving the screening task as being beyond their duties.

Other organisational barriers were inadequate guideline dissemination, insufficient planning, and lack of quality data monitoring systems (Table 4). For instance, Smith et al. [24] reviewed 10 countries at different stages of implementation of the World Health Organization Global Maternal Death Surveillance and Response and Maternal Death Review systems. They highlighted the lack of trained health workers to identify and collect data, poor quality of reporting, inaccurate reporting, poor data collection, use of handwritten reports (illegible and insufficient information, and lack of supervision or monitoring of reporting processes (weak registration systems) within services and the wider health system.

Several factors facilitated guideline adoption at the organisational level (Table 5). Targeted quality improvement initiatives, including well-planned implementation strategies, followed by periodic audit and monitoring of health professionals' guideline adherence were the most

widely reported initiatives by the studies (n = 17) (Table 5). Often organisations used distinct process models to inform practice change and guide implementation endeavours within the maternity services. For instance, systemic implementation science and research translation models [54] such as the Plan-Do-Study-Act (PDSA) cycles were used successfully to identify and implement evidence-based recommendations into healthcare and professional practice [38, 53, 55, 56]. Behaviour change models [54] such as the Theoretical Domains Framework [57] and the Theory of Change [26] specifically designed to address barriers and enablers to translating research into practice were also successfully applied to guide intervention activities and drive change effectively.

Ten studies highlighted organisational support, including championing change, the extent of stakeholder engagement in guideline development and implementation process as influential organisational factors for uptake of guidelines among health professionals. Several studies (n = 13) emphasised dissemination of information and promoting awareness of guideline content through training and education. Equally important were effective work design around guideline availability and decision support mechanisms for health professionals, including its integration into routine workflow at the point of care highlighted in 14 studies. Effective health data management systems and processes, including use of electronic data and health information management systems were also considered essential to effective guideline adoption [24, 25, 29, 45, 53].

## Health professional practice

Various factors impeded guideline adoption at the level of the health professional (clinicians) (Table 4). Fourteen studies highlighted lack of awareness, knowledge, and skills in relation to guidelines. For instance, a recent US-based study [32] assessed compliance of clinicians with the American Academy of Paediatrics (AAP) guideline on recommendations for use of intravenous immunoglobulin (IVIG) in infants with haemolytic disease. They found that lack of understanding of the possible side effects, cost, and donor exposure related to IVIG and lack of awareness of current recommendations for practice were potential barriers to compliance in neonatal intensive care units. Other clinician-related barriers included a lack of motivation and engagement by maternity unit staff, especially without incentives to attend training [22, 25, 45, 58, 66]. Initial resistance to change [22, 38, 46, 52, 66] and loss of autonomy were also cited barriers [38, 44] For example, Trollope et al. [44] noted that poor compliance with the national women's health maternity guidelines in New Zealand was associated with clinicians' perceptions that the guidelines did not reflect current evidence and failed to acknowledge patient individuality. Moreover, the clinicians believed that guideline recommendations would not lead to desired outcomes in care and may reduce their autonomy to make appropriate decisions about care [38].

Health workers' knowledge, attributes, beliefs, attitudes, and capability were prevalent barriers (Table 4). For example, there were differences in guideline adherence between disciplines. A UK-based study [59] examined nurses and doctors' knowledge and reported practice regarding procedural pain assessment and management in a neonatal intensive care unit. They found guideline adherence was more consistent among nurses than doctors. Differences were also reported in individual practice, such as variation in referral practices for management of gestational weight gain within maternity settings by different clinicians [35] and variation in use of labour management triage checklists by staff to decrease caesarean birth [38].

Health workers' knowledge and practice gaps were frequently reported (n = 16). For instance, a study assessing adherence of healthcare providers to hypertensive disorders of pregnancy guidelines in Jordan found nurses and midwives lacked sufficient knowledge about

pathophysiology of the disease and their role in monitoring mothers to reduce the risk of complications [41].

Other health professional practice challenges are related to some of the barriers previously highlighted in the organisational environment, such as lack of interdisciplinary communication and collaboration [25, 27]; quality of reporting and documentation [24, 25, 28, 60]; and lack of education and training about guidelines and updates [22–24, 30, 66]. Challenges of time constraints and heavy workload are common in maternal and neonatal healthcare and were a major barrier in the use of guidelines [35, 46, 60–62, 66].

Universally, the critical enablers for guideline uptake among health professionals were education, training, and feedback to staff on guideline adherence and most widely reported (n = 21) (Table 5). Other enablers for practice change and guideline adherence were ensuring clinicians' autonomy and ability to deviate from guidelines where person-centred care warranted change [44, 56]; interdisciplinary collaboration and communication [25, 27, 32, 36, 55, 60, 66]; and having established practice standards and expectations [27, 37, 42, 44, 45, 48, 52, 56, 60]. Additionally, more general factors playing an important part in supporting guideline adoption were related to health professionals' involvement in guideline implementation, and positive perceptions about guidelines usefulness, along with being able to effect behaviour change (Table 5).

## Discussion

This comprehensive review of 48 studies on the implementation of CPGs in maternity and neonatal care settings reveals a multifaceted landscape influenced by various factors across different levels of the healthcare system. Barriers and enablers centred around five main themes: (a) healthcare system factors; (b) patient and population factors; (c) guidelines and standards; (d) organisational capacity; and (e) health professional practice. These interrelated themes identified within maternity and neonatal care settings globally may also resonate in other healthcare settings.

Taking these themes together, enablers for guideline implementation and compliance may be considered as situated in the complex milieu of being: (i) supported at the political, economic and societal level; (ii) developed (and revised) with input from childbearing women, their support persons, parents of neonates and the wider community; (iii) aligned with nationally mandated clinical standards and consistent within a country; (iv) sufficiently resourced and accessible within maternity and neonatal services; and (v) used by clinicians with the appropriate mix of education, experience, expertise and motivation. It may be argued that the opposite of these ideals will present barriers to effective implementation of, or compliance with, CPGs.

Healthcare system factors and poorly resourced settings with inadequate governance or recognition of the needs of patients and staff were less likely to be able to effectively implement guidelines. Resources constraints were particularly problematic for low-and-middle income countries, subsequently affecting quality of care in maternal and neonatal settings [27]. Also, initiatives required by guidelines were not always affordable at the organisational or community level. Health literacy among women and their partners regarding interventions, along with individuals' beliefs, preferences and practices need to be considered in the development and implementation of guidelines to optimise effectiveness and compliance.

The effectiveness of CPG adoption was also impacted by the individual organisation's implementation processes (or the lack thereof). These organisational constraints are of major importance, therefore at the healthcare service level quality improvement initiatives and processes to support guideline implementation need to be prioritised at the healthcare service

level [7, 10]. As noted, a well-structured and supported implementation process benefited from tangible improvements and recognition of the efforts of staff.

Compliance with clinical guidelines may be impacted by (in)consistency between evidence, practice and interrelated guidelines within an individual maternity or neonatal service. Confidence or credibility and clarity in evidence is a critical factor influencing CPG adoption in healthcare, therefore guideline development should incorporate best-evidence available and relevance for the end users [4, 9]. One factor that warrants further exploration is the involvement of the woman, support person or parents in shared decision-making about care. Besides being central to effective management of care, shared-decision-making is considered as the desired outcome of CPG implementation [3]. Shared decision-making received some acknowledgement in reviewed articles in terms of guideline compliance: engagement with maternity services, health literacy level and self-efficacy were influenced by societal standards, opportunities, alternative faith-based care, and a requirement for approval from others. Despite the known benefits of continuity of (midwifery) care in improving shared decision-making, women's trust in guidelines has not featured in the deliberations about compliance [71].

Some evidence of a lack of credibility was noted, particularly when guidelines were poorly formulated. This may be explored further, particularly in the context of autonomy for healthcare professionals who may consider deviating from guidelines, however well-formulated and/or evidence-based, to provide an individualised, comprehensive approach to each mother and baby [56]. This would involve drawing on experience, expertise, guidelines, and evidence considered more appropriate in that context, ensuring that decisions are documented in the health record [9]. Also, ongoing education and interdisciplinary collaboration were crucial, while guideline accessibility and health information technology integration optimised adoption and these are widely supported broader CPG implementation strategies [5, 8].

In a 'perfect' setting, the enablers from each of the five interrelated and multidimensional themes (healthcare system, patient, guidelines and standards, organisation and health professional) would operate effectively and consistently, with barriers being addressed in the planning and delivery of the guideline implementation phase and as required over time. The reality is, of course, very different. So how might elements from these five themes be incorporated, while also considering the limitations, competing interests and priorities, and other complexities of the milieu in which they operate?

Future research, reflection and action are needed, to prioritise keeping the 'patient' (childbearing woman, support persons, parents, neonate) at the centre of clinical deliberations, including CPGs in shared decision-making [56, 72] and exploring the potential impact of continuity of care models. Other priorities include creating supportive healthcare environments that foster collaboration among and between clinician groups, and equip them with the necessary resources, knowledge and skills for effective guideline implementation. This may be explored for publicly and privately funded services, within and between discipline groups, for example, variable scope of practice for nurses, midwives, junior and more senior medical personnel, as well as within and between low-and-middle-income and high-income countries particularly in the context of their socio-political and economic backdrop and resource availability.

## Limitations

Our rapid review of barriers and enablers for implementing CPGs in the maternal and neonatal clinical setting, had a narrow focus, a short timeframe, and limited database search. However, a robust and rigorous review methodology drawing on appropriate guidance documents

were applied. Despite acknowledged limitations this review provides a comprehensive picture of the included studies and major contextual factors impacting CPG adoption. In considering these barriers and enablers, we acknowledge the authors' underlying assumption (and thus a potentially biased view) that CPGs should reflect "best practice" and optimise maternity and neonatal care.

## Conclusion

This review identified barriers and enablers for the implementation and adoption of clinical practice guidelines in maternal and neonatal settings at local and global levels. Organisational capacity, guidelines, patient factors, and health professional practices, underscore the complexity of guideline adoption across maternity and neonatal services. Addressing these complexities requires a comprehensive, collaborative strategy involving stakeholders at all levels. Ongoing evaluation, feedback mechanisms, and a commitment to evidence-based practices are crucial for successful and sustained adoption of clinical practice guidelines, ultimately contributing to improved maternal and neonatal outcomes in diverse healthcare settings.

## Supporting information

**S1 File. Full text screening summary.**
(DOCX)

**S2 File. Data extraction.**
(DOCX)

**S3 File. Expanded overview of study characteristics.**
(DOCX)

**S4 File. Quality appraisal.**
(DOCX)

**S5 File. Collated summary of key themes.**
(DOCX)

## Acknowledgments

The authors acknowledge the assistance of xx, PhD, who critiqued and edited the manuscript.

## Author Contributions

**Conceptualization:** Kalpana Raghunathan, Christine East.

**Data curation:** Kalpana Raghunathan.

**Formal analysis:** Kalpana Raghunathan, Christine East, Kritika Poudel.

**Methodology:** Kalpana Raghunathan, Christine East.

**Writing – original draft:** Kalpana Raghunathan, Christine East.

**Writing – review & editing:** Kalpana Raghunathan, Christine East, Kritika Poudel.

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
