## [Decision Letter · Decision Letter 0]

15 Oct 2024

PONE-D-24-16268Barriers and enablers for implementation of clinical practice guidelines in maternity and neonatal settings: A rapid reviewPLOS ONE

Dear Dr. Raghunathan,

Thank you for submitting your manuscript to PLOS ONE. After careful consideration, we feel that it has merit but does not fully meet PLOS ONE’s publication criteria as it currently stands. Therefore, we invite you to submit a revised version of the manuscript that addresses the points raised during the review process.

We look forward to receiving your revised manuscript.

Kind regards,

Vidanka Vasilevski

Academic Editor

PLOS ONE

Journal Requirements: When submitting your revision, we need you to address these additional requirements. 1. Please ensure that your manuscript meets PLOS ONE's style requirements, including those for file naming. The PLOS ONE style templates can be found at https://journals.plos.org/plosone/s/file?id=wjVg/PLOSOne_formatting_sample_main_body.pdf and https://journals.plos.org/plosone/s/file?id=ba62/PLOSOne_formatting_sample_title_authors_affiliations.pdf 2. Please identify your study as "systematic review" in the title of your manuscript. 3. As required by our policy on Data Availability, please ensure your manuscript or supplementary information includes the following:  A numbered table of all studies identified in the literature search, including those that were excluded from the analyses.   For every excluded study, the table should list the reason(s) for exclusion.   If any of the included studies are unpublished, include a link (URL) to the primary source or detailed information about how the content can be accessed.  A table of all data extracted from the primary research sources for the systematic review and/or meta-analysis. The table must include the following information for each study:  Name of data extractors and date of data extraction  Confirmation that the study was eligible to be included in the review.   All data extracted from each study for the reported systematic review and/or meta-analysis that would be needed to replicate your analyses.  If data or supporting information were obtained from another source (e.g. correspondence with the author of the original research article), please provide the source of data and dates on which the data/information were obtained by your research group.  If applicable for your analysis, a table showing the completed risk of bias and quality/certainty assessments for each study or outcome.  Please ensure this is provided for each domain or parameter assessed. For example, if you used the Cochrane risk-of-bias tool for randomized trials, provide answers to each of the signalling questions for each study. If you used GRADE to assess certainty of evidence, provide judgements about each of the quality of evidence factor. This should be provided for each outcome.   An explanation of how missing data were handled.  This information can be included in the main text, supplementary information, or relevant data repository. Please note that providing these underlying data is a requirement for publication in this journal, and if these data are not provided your manuscript might be rejected.  4. Thank you for stating the following financial disclosure: "The review activities were funded by Safer Care Victoria, Australia." Please state what role the funders took in the study.  If the funders had no role, please state: ""The funders had no role in study design, data collection and analysis, decision to publish, or preparation of the manuscript."" If this statement is not correct you must amend it as needed. Please include this amended Role of Funder statement in your cover letter; we will change the online submission form on your behalf.  5. Please include a separate caption for each figure in your manuscript. 6. Please include captions for your Supporting Information files at the end of your manuscript, and update any in-text citations to match accordingly. Please see our Supporting Information guidelines for more information: http://journals.plos.org/plosone/s/supporting-information. 7. Please review your reference list to ensure that it is complete and correct. If you have cited papers that have been retracted, please include the rationale for doing so in the manuscript text, or remove these references and replace them with relevant current references. Any changes to the reference list should be mentioned in the rebuttal letter that accompanies your revised manuscript. If you need to cite a retracted article, indicate the article’s retracted status in the References list and also include a citation and full reference for the retraction notice.

Reviewers' comments:

Reviewer's Responses to Questions

**Comments to the Author**

1. Is the manuscript technically sound, and do the data support the conclusions?

Reviewer #1: Yes

Reviewer #2: Yes

2. Has the statistical analysis been performed appropriately and rigorously? 

Reviewer #1: N/A

Reviewer #2: N/A

3. Have the authors made all data underlying the findings in their manuscript fully available?

Reviewer #1: Yes

Reviewer #2: Yes

4. Is the manuscript presented in an intelligible fashion and written in standard English?

Reviewer #1: Yes

Reviewer #2: Yes

5. Review Comments to the Author

Reviewer #1: Thank you for the opportunity to review this manuscript. The subject matter is relevant and I read with interest. The manuscript is written with clarity, good flow and structure and includes the necessary detail. Overall this is an interesting review and provides an insight into the complexities and considerations in the maternity and neonatal context.

It would add value and clarity for the reader to add a description of the healthcare staff typically involved in maternity and neonatal care. Describing the healthcare professions that have a role to play in implementing these CPG would enhance standardising the language used described in the themes.

In addition a description of what constitutes maternity and neonatal care in the countries where the literature was sought would further provide clarity for the international audience.

Specifically some minor amendments:

Line 313, 314 add reference for World Health Organization Global Maternal and Death Surveillance and Response and Maternal Death Review systems

Line 316/317 review wording and ()

Line 330 and 332 Consider replacing ‘Many’ and ‘Several’ with a more objective term, eg a number of studies, or some studies

Table 4: review wording of title

Thank you

Reviewer #2: This manuscript is technically sound requiring minor revisions for clarity. This research contributes new evidence to the field of knowledge translation. Thematic analysis has been described adequately. Clarification of some health care factors in Table 4 has been requested, otherwise the findings are clearly presented. Further editing is needed to correct several spelling errors and consistency with referencing (Table 5). Please see attached file for additional comments.

6. PLOS authors have the option to publish the peer review history of their article (what does this mean?). If published, this will include your full peer review and any attached files.

Reviewer #1: No

Reviewer #2: **Yes: **Melissa Blake

---

## [Author Response · Author response to Decision Letter 0]

26 Nov 2024

Responses provided in submitted Response to Reviewer document

---

## [Editor Report · Decision Letter 1]

28 Nov 2024

Barriers and enablers for implementation of clinical practice guidelines in maternity and neonatal settings: A systematic rapid review

PONE-D-24-16268R1

Dear Dr. Raghunathan,

We’re pleased to inform you that your manuscript has been judged scientifically suitable for publication and will be formally accepted for publication once it meets all outstanding technical requirements.

Kind regards,

Vidanka Vasilevski

Academic Editor

PLOS ONE

Additional Editor Comments (optional):

You have addressed the author comments appropriately. I agree with the authors that using the term 'systematic' in the title may be misleading, and as such could be removed from the paper. This can be addressed at the copy-editing phase of publication.

---

## [Editor Report · Acceptance letter]

2 Dec 2024

PONE-D-24-16268R1 

PLOS ONE

Dear Dr. Raghunathan, 

I'm pleased to inform you that your manuscript has been deemed suitable for publication in PLOS ONE. Congratulations! Your manuscript is now being handed over to our production team.

Kind regards, 

on behalf of

Dr. Vidanka Vasilevski 

Academic Editor

PLOS ONE